# Optimization of Extraction Process and Dynamic Changes in Triterpenoids of *Lactuca indica* from Different Medicinal Parts and Growth Periods

**DOI:** 10.3390/molecules28083345

**Published:** 2023-04-10

**Authors:** Junfeng Hao, Qiang Si, Zhijun Wang, Yushan Jia, Zhihui Fu, Muqier Zhao, Andreas Wilkes, Gentu Ge

**Affiliations:** 1College of Grassland Resources and Environment, Inner Mongolia Agricultural University, Hohhot 010019, China; 2Key Laboratory of Forage Cultivation and the Processing and High Efficient Utilization of the Ministry of Agriculture, Inner Mongolia Agricultural University, Hohhot 010019, China; 3Key Laboratory of Grassland Resources of the Ministry of Education, Inner Mongolia Agricultural University, Hohhot 010019, China; 4Values for Development Limited, Cambridge CB4 1RS, UK

**Keywords:** *Lactuca indica* L.cv. Mengzao, triterpenoids, microwave extraction, growth period, oxidation resistance

## Abstract

In this study, the triterpenoids in the leaves of *Lactuca indica* L.cv. Mengzao (LIM) were extracted via microwave-assisted ethanol extraction, and the optimum extraction conditions for triterpenoids were determined through single-factor experiments and the Box–Behnken method. The effects of three factors (solid–liquid ratio, microwave power and extraction time) on the total triterpenoids content (TTC) were evaluated. The TTC of different parts (roots, stems, leaves and flowers) of LIM in different growth stages was studied, and the scavenging effects of the highest TTC parts on DPPH, ABTS and hydroxyl free radicals were investigated. The results showed that the optimum extraction conditions for microwave-assisted extraction of total triterpenoids from LIM leaves were as follows: solid–liquid ratio of 1:20 g/mL; microwave power of 400 W; and extraction time of 60 min. Under these conditions, the TTC was 29.17 mg/g. Compared with the fresh raw materials, the TTC of the materials increased after freeze drying. The leaves of LIM had the highest TTC, and the flowering stage was the best time. The triterpenoids from the leaves had a strong ability to eliminate DPPH and ABTS free radicals, and the elimination effect of dried leaves was better than that of fresh leaves, while the elimination effect of hydroxyl free radicals was not obvious. The tested method was used to extract total triterpenoids from LIM using a simple process at low cost, which provides a reference for developing intensive processing methods for *L. indica*.

## 1. Introduction

*Lactuca indica* L.cv. Mengzao (LIM) is an annual plant of the Compositae family. All parts of *Lactuca indica*, including the roots, stems, leaves, flowers and seeds, have been used as traditional Chinese medicines for thousands of years [1]. It has been widely used as a kitchen herb for salads, sushi and soup, due to its delicate leaves and high nutritional value [2]. It is fresh, juicy, slightly bitter, rich in flavonoids, lipids, terpenes, sterols and other substances, and has biological effects such as heat clearing and detoxification, anti-aging and anti-tumor effects, and anti-inflammatory, analgesic and antioxidant properties. Among the plant contents, triterpenoids are compounds widely distributed in plants, and have attracted increasing interest because of their antioxidant activity [3,4]. The main triterpenoid compounds in *L. indica* are oleanolic-acid-type triterpenoids [5]. Triterpenoids are compounds formed by the condensation of six isoprene monomers, which are important natural chemical components. To date, most of the triterpenoids found have been tetracyclic triterpenoids and pentacyclic triterpenoids, which have many effects such as tumor inhibition, liver protection and immunity enhancement [3,6].

Several studies have been conducted on extraction methods for total triterpenoids in *L. indica*. Traditional extraction methods include reflux extraction [7] and Soxhlet extraction [8], which are associated with long drying times, high energy consumption and low efficiency. With the development of the concept of “green extraction” [9], various new extraction technologies are increasingly being used to extract natural products from plants, including microwave-assisted extraction [10], ultrasound-assisted extraction [11] and supercritical fluid extraction [12]. Among these extraction technologies, microwave extraction is relatively popular and has some obvious advantages, e.g., low energy consumption, short extraction time, higher extraction rate and lower cost [13]. To date, there have been few reports on the extraction process and antioxidant activity of total triterpenoids from *L. indica*, and also few reports on the distribution of total triterpenoids in different parts of the plant and the best harvest time.

In this study, the raw material was LIM, a nationally approved variety independently cultivated and bred by the team. On the basis of single-factor experiments, the effects of the solid–liquid ratio, microwave power and extraction time on total triterpenoids in LIM leaves were investigated to determine the best extraction conditions. On this basis, the distribution of and dynamic changes in the TTC in various parts (roots, stems, leaves and flowers) of LIM in different growth periods were investigated, the optimal utilization part and harvest time were determined, and its abilities to eliminate DPPH free radicals, hydroxyl free radicals and ABTS free radicals were investigated. It is expected that this research will provide a theoretical basis and reference for further development of diversified and intensive processing of *L. indica*.

## 2. Results and Discussion

### 2.1. Optimization of TTC Conditions

#### 2.1.1. Influence of Solid–Liquid Ratio on TTC

The solid–liquid ratio is a key factor for an efficient recovery of biomolecules from different materials, since the extraction of these compounds is influenced by the mass transfer resistance associated with the solid matrix structure. Moreover, in the face of a subsequent industrial scale-up, a compromise must be reached between a sufficient solid–liquid ratio to achieve a suitable extraction yield that is low enough to reduce the energy costs [14]. In this study, as the solid–liquid ratio increased from 1:5 to 1:20 g/mL, the TTC increased sharply (Figure 1a). When the solid–liquid ratio increases, the cavitation effect generated by microwaves will be stronger, and the mass transfer process will be faster, resulting in a higher extraction efficiency [15]. However, when the solid–liquid ratio increased to 1:20 g/mL, the effective components were basically dissolved by a certain proportion of the solvent, so the extraction rate stopped increasing. Furthermore, if the amount of ethanol is too high, the dissolution of other impurities will also increase correspondingly, resulting in a decrease in the extraction volume of triterpenoids [16]. Therefore, from the perspective of the extraction volume, solvent consumption and production cost, a solid–liquid ratio of 1:15~1:25 g/mL was selected for further optimization.

#### 2.1.2. Influence of Microwave Power on TTC

Microwave-assisted extraction is one of the most commonly used extraction methods. Microwave heating can improve the diffusion ability of target components, thereby shortening extraction time [17]. This showed that the microwave power has a certain impact on the TTC (Figure 1b). The TTC reached a maximum at 400 W, and then showed a downward trend. This result is consistent with previous reports also showing that the TTC increased with increasing microwave power within a certain microwave power range [18,19]. When the microwave power is too high, the solvent volatilizes. Therefore, a microwave power of 300~500 W was selected for further optimization.

#### 2.1.3. The Influence of Extraction Time on TTC

The choice of extraction time was another important step to guarantee the distribution equilibrium of triterpenoids between the sample and extraction solvent [20]. The results are shown in Figure 1c. As the extraction time increased from 15 min to 60 min, the TTC increased significantly. After 60 min, the TTC began to decline. With a longer extraction duration, the TTC increased first and then decreased. The reason may be that at the beginning of extraction, the extraction volume increased due to the different concentrations of raw materials and solvents. After a certain duration, the mass concentration of active substances inside and outside the raw material reached a relative balance, and the triterpenoids in the raw material were no longer easily leached, so the extraction volume tended to stabilize. In addition, the extraction of triterpenoids will be affected by the dissolution of other alcohol-soluble substances during the extraction period [21], resulting in leaching of impurities, thus reducing the TTC. After comprehensive consideration, 45~75 min was selected for further optimization.

### 2.2. Optimization of Microwave-Assisted Parameters for Triterpenoids by BBD

#### 2.2.1. Model Fitting Analysis

After these pre-experiments, three main variable levels were determined for the solid–liquid ratio (1:15~1:25 g/mL), microwave power (300~500 W) and extraction time (45~75 min). The experimental design and corresponding response data for the TTC from LIM are presented in Table 1. Regression analysis showed that the extraction of the TTC was predicted by the second-order equation (Equation (1))
Y = 32.94 + 0.28A + 0.60B − 0.39C + 0.77AB + 1.24AC − 0.40BC − 2.05A^2^ − 2.72B^2^ − 2.05C^2^(1)
where Y represents the response of the TTC, A represents the solid–liquid ratio, B represents the microwave power and C represents the extraction time.

Table 2 shows the results of calculating the regression coefficients for the intercept, and linear, quadratic and interactive terms of the model. Significance testing (*p* < 0.001) of the regression equations showed that the quadratic multiple regression model was extremely significant. The independent variables (A, B, C) and the three quadratic terms (A^2^, B^2^, C^2^) of the equation had a significant impact on the TTC (*p* < 0.05). There was also a significant interaction between the solid–liquid ratio and microwave power (AB), and between the solid–liquid ratio and extraction time (AC) (*p* < 0.05). For each item in the model, a larger F value and a smaller *p* value indicate a more significant impact on the corresponding response variables [22]. From the results of the variance analysis in Table 2, the decision coefficient (R^2^) of the model was 0.98, and the adjusted decision coefficient (Adj-R^2^) was 0.96, indicating that the prediction results had a strong correlation with the actual results [23]. The model *p* value was less than 0.05, indicating that the experimental model was significant. The mismatch was not significant (*p* = 0.1129 > 0.05), indicating that the model fit the actual data well. The F values showed that the order of influence of various factors on the TTC was as follows: microwave power (B) > extraction time (C) > solid–liquid ratio (A).

#### 2.2.2. Model Applicability Diagnosis

The applicability of the linear regression simulation of the total triterpenoids extracted from the leaves of LIM was diagnosed using the three diagnostic charts for the residual normal probability distribution, the studentized residuals and the predicted and measured values. The results are shown in Figure 2. It can be seen from the residual normal probability distribution diagram in Figure 2a that the distribution of most data points was concentrated together, and was close to a straight line, indicating that the model was stable and consistent with a normal distribution. According to the studentized residual diagram in Figure 2b, all data points were distributed within the range of ±3, indicating that the model was in good agreement with the total triterpenoids extracted from LIM. It can be seen from Figure 2c that the data points of the predicted values and the actual values were close to a straight line, and the model fit well across the whole regression area. The model can effectively predict the total triterpenoid extraction amount in the leaves of LIM [10].

#### 2.2.3. Response Surface Optimization Analysis of Triterpenoid Extraction Conditions

The influence of variables and their interactions on the TTC is illustrated through an inspection of the three-dimensional (3D) response surfaces and two-dimensional (2D) contour maps (Figure 3a–f). The interaction between variables can be interpreted from the shape of the contour map. An elliptic contour map shows that the interaction between the corresponding variables was significant, while a circular contour map shows that the interaction between the corresponding variables was not significant [24]. The maximum predicted response was located at the peak of the three-dimensional response surface, and the corresponding point was defined by the minimum ellipse in the two-dimensional contour map. The steepness of the surface can reflect the impact of the investigated factors on the response value. The greater the steepness, the greater the impact on the response value [25]. In this study, the opening of the 3D response surface was downward. With an increase in two factor values (AB, AC, BC), the response value of the TTC will also increase. After the response value increased to the extreme value, it gradually decreased with the increase in two factor values.

Figure 3a,b show the comprehensive effects of the solid–liquid ratio and microwave power on the TTC. It is clearly shown that when the solid–liquid ratio was increased to 1:20 g/mL and the microwave power increased to 400 W, the TTC increased and reached its peak. As the two factors further increased, the TTC gradually decreased. The results showed that only a proper solid–liquid ratio and a certain extraction power could significantly improve the TTC, and too low or too high values of these factors would lead to a decrease in extraction yield. In addition, the contour map was elliptical, indicating that the interaction between the ultrasonic power and extraction time was significant [26].

Figure 3c,d show the comprehensive effect of the solid–liquid ratio and extraction time on the TTC. When the microwave power remained unchanged, the TTC increased with increasing solid–liquid ratio (1:15~1:20 g/mL) and extraction time (45~60 min), and reached a peak in the middle stage. As the solid–liquid ratio and extraction time increased, the TTC began to decrease, which may be due to the dissolution of other alcohol-soluble substances after a long extraction period in the presence of more solvents. In addition, the elliptical shape of the contour map indicated a significant interaction between these two factors.

Figure 3e,f show the combined effect of the microwave power and extraction time on the TTC. Increasing the microwave power to 400 W led to an increase in the TTC, but a further increase in the microwave power led to a slow decrease in the TTC. For the effect of the extraction time on the TTC, the TTC reached its maximum after about 60 min and then decreased slightly, which may be due to the decomposition of triterpenoids with the increased extraction time. The circular contour map shows that the interaction between these two parameters was not significant.

The results of the optimization using Design-Expert 8.0.6 analysis software showed that the best extraction conditions for the TTC were a solid–liquid ratio of 1:20.33 g/mL, microwave power of 412.44 W and extraction time of 58.75 min. Under this condition, the predicted value of the TTC in LIM leaves was 29.05 mg/g. In order to ensure that the predicted value did not deviate from the actual results, the best conditions derived were verified through experiments. Based on the operability of the test, the best parameters for prediction were fine-tuned, i.e., solid–liquid ratio of 1:20 g/mL, microwave power of 400 W and extraction time of 60 min. Three parallel validation experiments were conducted under this condition, and the average extraction yield of total triterpenoids was 29.17 ± 0.15 mg/g, which verified the accuracy of the RSM model. The reliability of the optimized process parameters was fully explained.

### 2.3. Dynamic Changes in TTC in Different Parts of LIM in Different Growth Stages

The TTC extraction test was applied to fresh and dried samples of different parts of LIM harvested in different growth stages using the optimized extraction process. The TTC of the roots, stems, leaves, flowers and the whole plant of LIM harvested in different growth stages was measured three times in parallel, as shown in Figure 4. From the perspective of sample treatments, compared with the fresh raw materials, the TTC of the whole plant and of each plant part increased after freeze-drying. Among the different parts, the increase in the TTC in the leaves and flowers was the most obvious. Due to the high moisture content of the fresh samples and low TTC, the TTC of each part showed an increasing trend with the loss of moisture during the drying process [27]. In addition, triterpenoids are secondary metabolites of plants, which are usually the products of environmental stresses such as water and temperature, and the drying process is a drought stress process for freshly harvested plants, which may induce the formation of triterpenoids and an increase in the relevant active components [28]. In this study, the TTC of each part of LIM increased significantly after drying, which may be due to the formation or increased content of triterpenoids induced by drought stress.

The content of active substances in *L. indica* varies during the plant growth process. Dong [29] pointed out that the TTC in *L. indica* showed a downward trend before flowering, and then increased sharply to reach a maximum value. The research results of Wang [30] showed that the seedling stage was the harvesting stage with the highest TTC in *L. indica*. In this experiment, the total triterpenoids were extracted from fresh samples of LIM in different periods. The results showed that the TTC in the initial flowering stage was the highest, after which a downward trend was observed. The freeze-dried plants also returned similar results, indicating that the initial flowering period is the most suitable period for the extraction of total triterpenoids in LIM. In addition, the TTC of different parts of LIM showed dynamic changes during the growth process.

Figure 4e,f show that the leaves had the highest TTC in all the fresh and dried samples. Figure 4E shows that the extraction yield of fresh leaf samples in the middle flowering stage was the highest, reaching as high as 19.19 mg/g. The yield of dried leaf samples extracted in the middle flowering stage was the highest, reaching 33.96 mg/g. Among the fresh samples, the extraction yield of the roots in the seeding stage was the lowest, at 1.82 mg/g. Among the dried samples, the extraction yield of the samples extracted from the whole plants in the filling stage was the lowest, at 5.39 mg/g. As the main organ for transporting nutrients, the stems provide a pathway for the synthesis and transportation of total triterpenoids to the roots, leaves and flowers [4]. Leaves, as the main organ for the accumulation of effective components, accumulate a relatively high TTC. Therefore, leaves should be considered the main plant part targeted for the extraction of triterpenoids in LIM. In order to study the antioxidant activity of the total triterpenoids of LIM, fresh and dried leaf samples in the middle flowering stage were selected to test their antioxidant activity.

### 2.4. Antioxidant Activity of Total Triterpenoids in LIM

DPPH is a purple stable free radical. It can absorb light at 517 nm. After receiving the hydrogen atom of an antioxidant, it changes from purple to yellow. After adding a certain antioxidant, some free radicals are removed, and the absorption intensity at this wavelength is weakened, so it is often used to evaluate the antioxidant properties of substances [31]. In this study, taking L-ascorbic acid as the positive control, the total triterpenoids in the leaves of LIM showed a strong scavenging effect on DPPH, and the scavenging ability was positively correlated with the concentration of total triterpenoids (Figure 5a). In the concentration range of 0.1~0.6 mg/mL, the scavenging ability of DPPH increased significantly with increasing total triterpenoid concentration in the leaves of LIM. The scavenging effect of fresh and dried leaves of LIM on DPPH was lower than that of L-ascorbic acid, and their IC_50_ values were 0.21 mg/mL and 0.28 mg/mL, respectively, indicating that drying fresh leaves of LIM can enhance the scavenging effect on DPPH free radicals.

Figure 5b shows that L-ascorbic acid, dried leaves and fresh leaves of LIM with different mass concentrations had a certain elimination effect on ABTS^+^ free radicals. In the concentration range of 0.1~0.6 mg/mL, the scavenging effect of dried leaves and fresh leaves on ABTS free radicals increased with increasing TTC in the extracts. The scavenging effect of 0.6 mg/mL was the strongest in the dried and fresh leaves, with scavenging rates of 72.99% and 55.51%, respectively, which were weaker than those of L-ascorbic acid, and their IC_50_ values were 0.18 mg/mL and 0.32 mg/mL, respectively. These results showed that drying fresh leaves of LIM enhanced the TTC scavenging effect on ABTS^+^ free radicals.

A reaction system model was established according to the Fenton reaction method, and the mixture of H_2_O_2_ and Fe^2+^ was used to generate hydroxyl. Then, salicylic acid was added to produce colored products. The product has strong absorption at the 510 mm wavelength. If an antioxidant is added to the reaction system, it will compete with salicylic acid for hydroxyl and reduce the amount of colored products [32]. In this study, a series of extracts of dried and fresh leaves of LIM with different concentrations were added to the reaction system. By measuring the absorbance of each concentration at the 510 nm wavelength, the scavenging effects of the tested substances on hydroxyl radicals were detected. Figure 5c shows that the total triterpenoids in the fresh and dried leaves increased the scavenging effect on hydroxyl radicals as the concentration of the extract increased. The scavenging effect of both fresh and dried leaves on hydroxyl radicals was weaker than that of L-ascorbic acid, and their IC_50_ values were 0.36 mg/mL and 0.74 mg/mL, respectively. This showed that drying fresh leaves of LIM enhanced the scavenging effect of the leaves on ABTS free radicals.

## 3. Materials and Methods

### 3.1. Plant Materials

LIM samples were collected from the planting base of Inner Mongolia Agricultural University from June to September 2019. The collected samples were divided into different parts, i.e., roots, leaves, stems and flowers. All samples were prepared in two ways, fresh and dried. The fresh samples were processed for 2 min by crushing the sample with an RCD-1A homogenizer (Jiangsu Dongpeng Instrument Manufacturing Co., Ltd., Changzhou, China), while the dried samples were vacuum freeze-dried for 48 h using a Christ ALPHA 1-4LDplus lyophilizer (Marin Christ, Osterode, Germany) and then crushed into a uniform powder (40 mesh) for analysis.

### 3.2. Chemicals and Reagents

Oleanolic acid, L-ascorbic acid, 1,1-diphenyl-2-trinitrophenylhydrazine (DPPH) and 2,2′-diazo-bis-3-ethylbenzothiazoline-6-sulfonic acid (ABTS) were all purchased from Shanghai Yuanye Technology Co. Ltd. (Shanghai, China). Hydrogen peroxide, ferrous sulfate, salicylic acid and ethanol were all purchased from Sinopharm Chemical Reagent Co. Ltd. (Shanghai, China). All chemicals and reagents were of analytical grade.

### 3.3. Microwave-Assisted Extraction of Total Triterpenoids

Total triterpenoid extraction was performed using an XH-100B series microwave extraction instrument (Beijing Xianghu Science and Technology Development Co. Ltd., Beijing, China). Freeze-dried samples (crushed through a 40-mesh sieve) and fresh sample fragments (cut to a particle size of about 1 mm) of LIM leaves were added into a conical flask with a stopper, and 70% ethanol solution [30] was added to achieve the target ratio of material to liquid. The sample solution was extracted under different microwave powers and treatment times before centrifuging (15,000 rpm, 15 min) and extracting the supernatant. The obtained solution was diluted to 50 mL with 50% ethanol solution by volume fraction, and then shaken to obtain the triterpenoid extracts from the leaves of LIM. Samples were stored at 4 °C. Each experimental treatment was repeated three times.

### 3.4. Determination of TTC 

An amount of 25 mg of oleanolic acid was weighed and fully dried at 105 °C. The solution was dissolved and diluted with methanol into a 25 mL volumetric flask, shaken to make it completely dissolve and prepared as a reference solution with a mass concentration of 1.0 mg/mL. Amounts of 0, 0.2, 0.4, 0.6, 0.8 and 1 mL of the oleanolic acid reference solution with a mass concentration of 1.0 mg/mL were placed into different test tubes. After completely volatilizing the solvent at room temperature, 1.6 mL of perchloric acid solution and 0.4 mL of 5% vanillin–acetic acid solution were added to each test tube, heated in a 70 °C water bath for 15 min and cooled naturally, and then 5 mL of glacial acetic acid was added and shaken well. The absorbance of the reference solution of different concentrations at the wavelength of 548 nm was determined with a UV-1900i series ultraviolet–visible spectrophotometer (Shimadzu Co. Ltd., Kyoto, Japan), and a blank test was performed. With different concentrations of oleanolic acid as the abscissa and absorbance A as the ordinate, we drew the standard curve. The standard curve equation was y = 0.0862x + 0.0012, R^2^ = 0.9999.

An amount of 1 mL of total triterpene extracts of LIM was taken, and the concentration of total triterpenes in the LIM sample solution was measured according to the method of the standard curve equation. 

### 3.5. Single-Factor Experiment

The effects of the solid–liquid ratio, microwave power and extraction time on the TTC from the leaves of LIM were studied by changing the level of one factor and keeping the other two factors constant. The detailed conditions for each extraction were as follows: (1) when the liquid–solid ratios were 5, 10, 15, 20 and 25 g/mL, the samples were extracted with an ethanol concentration of 70% at 300 W for 30 min; (2) when the microwave power settings were 200, 300, 400, 500 and 600 W, the samples were extracted with a liquid–solid ratio of 10 g/mL and an ethanol concentration of 70% for 30 min; (3) when the extraction times were 15, 30, 45, 60 and 75 min, the samples were extracted with a solid–liquid ratio of 10 g/mL and an ethanol concentration of 70% at 300 W.

### 3.6. Optimization Experimental Design

Based on the preliminary results of the single-factor test, the proper range for each factor was determined, and then response surface methodology (RSM) was conducted using BBD in Design Expert 8.0.6 software (StatEase^®^, Minneapolis, MN, USA). As shown in Table 3, the three factors chosen for this study were designed as A (solid–liquid ratio), B (microwave power) and C (extraction time) and were prescribed three levels, coded as −1, 0 and 1. The TTC was the response value. The experimental design included 19 trials, including 5 replicates of the center point.

### 3.7. Determination of Antioxidant Activity of MAE Extracts

#### 3.7.1. DPPH Radical Scavenging Assay

The DPPH radical scavenging assay of the sample extracts was performed as described in a previous study with slight modifications [33]. An amount of 2 mL of sample extraction solution with different mass concentrations (0.1~0.6 mg/mL) was mixed with 2 mL of 0.1 mg/mL freshly prepared DPPH solution (in methanol). After incubation in the dark for 30 min at room temperature, the absorbance (A_i_) at 517 nm was read against a blank using a UV-1900i series ultraviolet–visible spectrophotometer. The DPPH scavenging activity was calculated using the formula
Elimination rate(%) = [1 − (A_i_ − A_j_)/A_O_] × 100%(2)
where Ao is the absorbance of the control group (70% ethanol solution instead of sample solution), and A_j_ is the absorbance of the negative control (methanol instead of DPPH solution). L-ascorbic acid (0.1~0.6 mg/mL) was used as the reference. The semi-inhibitory concentration (IC_50_) value, the sample concentration required to quench 50% of DPPH radicals, was obtained by fitting the polynomial using Statistical Product and Service Solutions 14.0 (SPSS Inc., Chicago, IL, USA). The lower the IC_50_ value, the stronger the free radical scavenging ability of the antioxidant.

#### 3.7.2. ABTS Radical Scavenging Assay

The ABTS radical scavenging assay applied to the sample extracts was performed as described in a previous study with slight modifications [34]. With L-ascorbic acid as the positive control, the absorbance of the sample extracts with different concentrations (0.1~0.6 mg/mL) was determined at 734 nm using an ultraviolet spectrophotometer. The ABTS radical scavenging activity was calculated according to Equation (2), where Ao represents the absorbance of the control group (70% ethanol solution instead of sample solution), A_i_ represents the absorbance of the sample solution and A_j_ represents the absorbance of the sample only (methanol instead of ABTS solution).

#### 3.7.3. Hydroxyl Radical Scavenging Activity

The hydroxyl radical scavenging assay for the sample extracts was performed as described in a previous study with slight modifications [35]. An amount of 2 mL of 6 mmol/L FeSO_4_ solution was mixed with 2 mL of extraction solution with different mass concentrations (0.1~0.6 mg/mL), and 2 mL of 6 mmol/L H_2_O_2_ solution was added to the mixed solution and left to stand for 10 min. Then, 2 mL of 6 mmol/L salicylic acid solution was added and left to stand for 30 min. The hydroxyl radical scavenging rate was computed according to Equation (2), where A_i_, A_j_ and Ao correspond to the absorbances of the sample solution reaction, reagent blank (without DPPH solution) and control (without the sample), respectively, all of which were measured at 510 nm.

### 3.8. Statistical Analysis

Data are expressed as the mean ± standard deviation of triplicate measurements. Analysis of variance was performed using SPSS 24.0 (SPSS Inc., Chicago, IL, USA). Statistical analysis of the results was carried out using analysis of variance (ANOVA) with Duncan’s test, and the statistical significance of differences between the means of each factor was determined at *p* < 0.05. Charts were drawn with OriginPro 2021 (Origin Lab^®^, Northampton, MA, USA).

## 4. Conclusions

Our method provides information on the complete extraction and antioxidant determination of triterpenoids in LIM. In this study, three-variable, three-stage experiments conducted using BBD and RSM were applied to investigate the microwave-assisted ethanol aqueous solution extraction method to determine the highest extraction rate of triterpenoids in LIM leaves. Furthermore, the differences in total triterpenoids in different parts of LIM harvested in different growth periods were analyzed. These data not only show that there were differences in the active components of the aboveground and underground parts of LIM, but also provide an effective reference for the determination of the best harvest time of different plant parts. The leaves are the main part to harvest, and the flowering stage is the best harvest time. Our study also shows that the total triterpenoids in the leaves have a high antioxidant potential in vitro, with potential applications in food, feed and nutrients. In the future, in vitro digestibility, bioavailability, toxicology and animal research is needed to develop leaves into commercial ingredients.

## Figures and Tables

**Figure 1 molecules-28-03345-f001:**
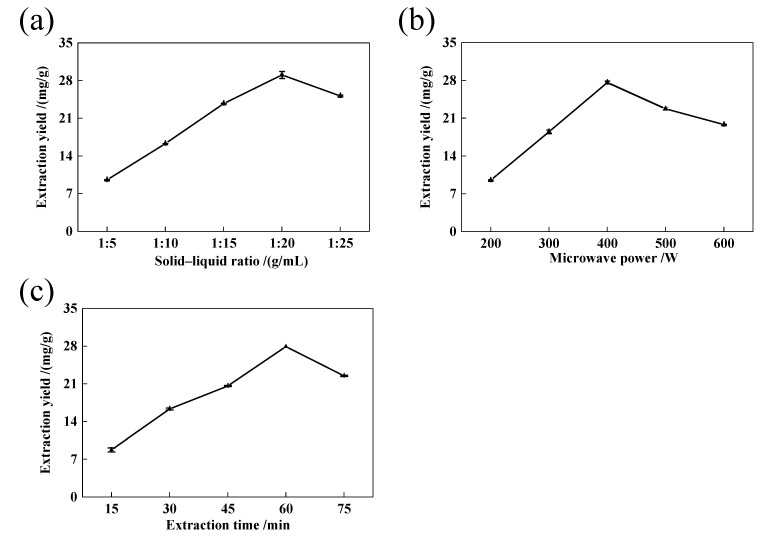
The effects of the solid–liquid ratio (**a**), microwave power (**b**) and extraction time (**c**) on TTC extraction from LIM leaves.

**Figure 2 molecules-28-03345-f002:**
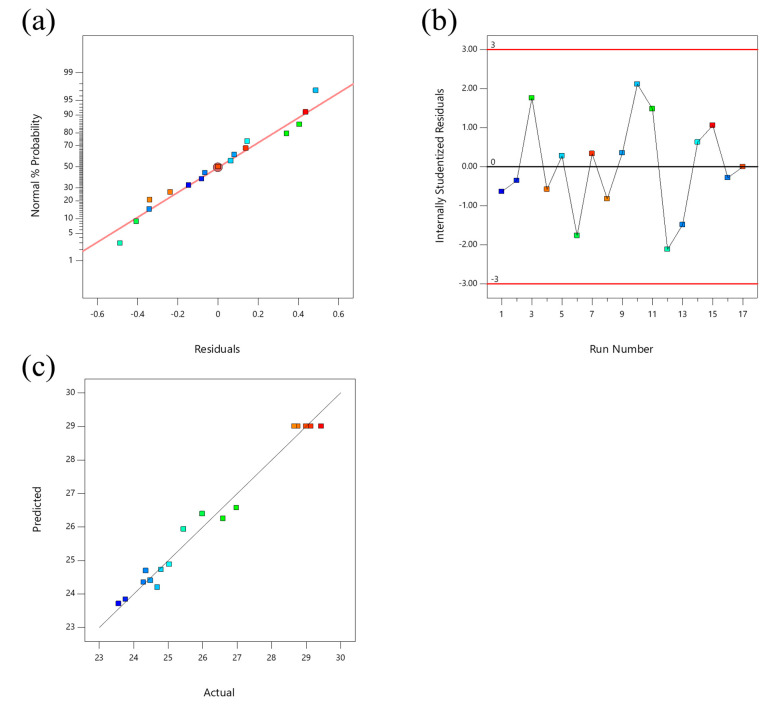
Diagnostic plots for model adequacy for extraction of TTC. (**a**) Residual normal probability plot; (**b**) studentized residuals; (**c**) predicted and actual values. The data points with different colors in the figure represent the test groups with different TTCs.

**Figure 3 molecules-28-03345-f003:**
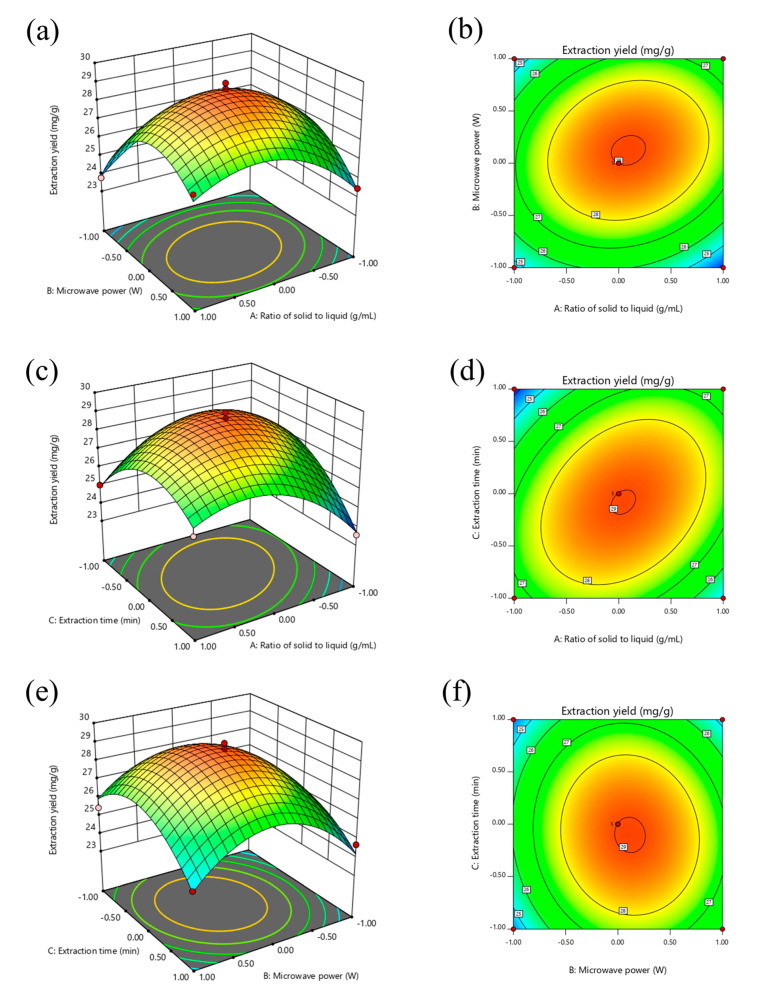
Three-dimensional response surface plots showing the effects of different parameters on TTC. Solid–liquid ratio and ultrasonic power (**a**); solid–liquid ratio and ethanol concentration (**b**); solid–liquid ratio and extraction time (**c**); ultrasonic power and ethanol concentration (**d**); ultrasonic power and extraction time (**e**); ethanol concentration and extraction time (**f**).

**Figure 4 molecules-28-03345-f004:**
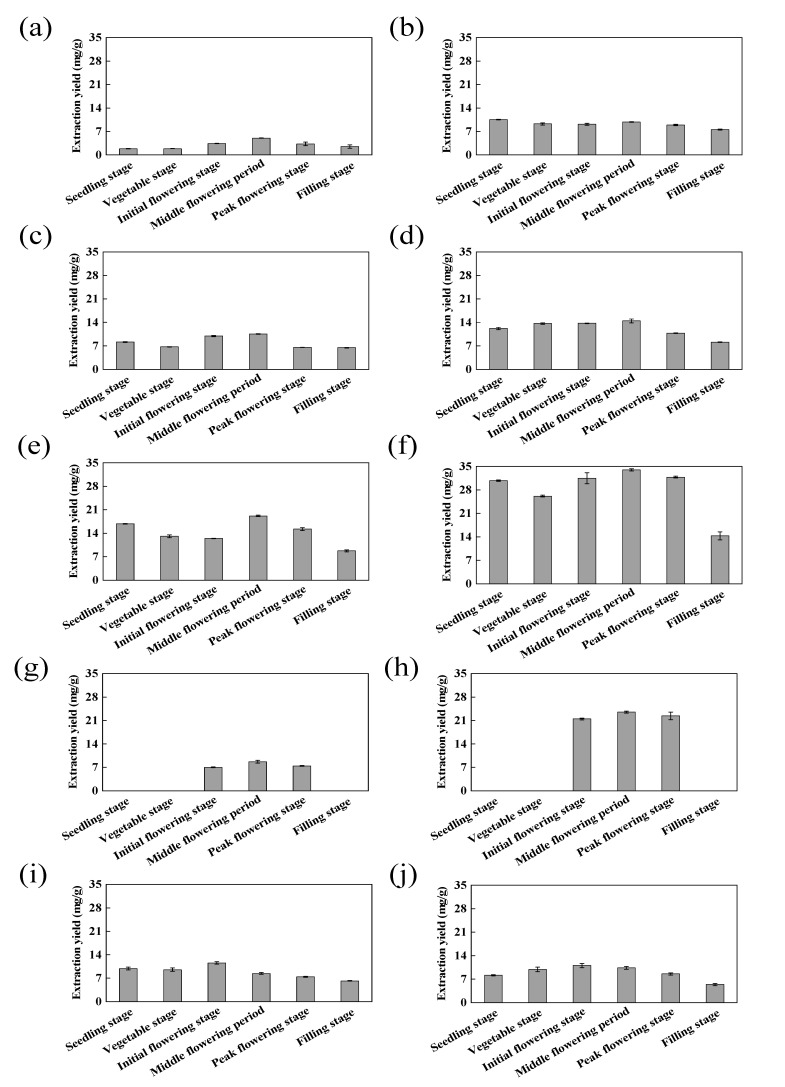
Analysis of TTC from fresh *Lactuca indica* samples (**a**)—roots; (**c**)—stems; (**e**)—leaves; (**g**)—flowers; (**i**)—whole plants) and dried *Lactuca indica* samples (**b**)—roots; (**d**)—stems; (**f**)—leaves; (**h**)—flowers; (**j**)—whole plants).

**Figure 5 molecules-28-03345-f005:**
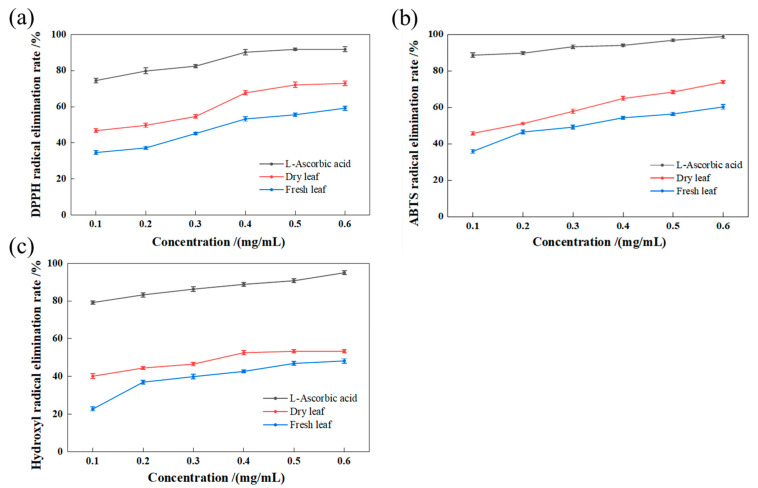
Antioxidant activities (**a**)—DPPH radical elimination ability; (**b**)—ABTS radical elimination ability; (**c**)—hydroxyl radical elimination ability) of total triterpenoids in fresh and dried leaves of LIM.

**Table 1 molecules-28-03345-t001:** Box–Behnken program and experimental results.

Test	Solid–Liquid Ratio	Microwave Power	Extraction Time	Response of TTC
	/(g/mL)	/(W)	/(min)	/(mg/g)
1	1:15	300	60	27.66
2	1:25	300	60	26.99
3	1:15	500	60	27.81
4	1:25	500	60	30.20
5	1:15	400	45	30.64
6	1:25	400	45	28.43
7	1:15	400	75	26.76
8	1:25	400	75	29.52
9	1:20	300	45	27.58
10	1:20	500	45	28.90
11	1:20	300	75	28.03
12	1:20	500	75	28.16
13	1:20	400	60	32.67
14	1:20	400	60	32.55
15	1:20	400	60	33.10
16	1:20	400	60	32.94
17	1:20	400	60	33.44

**Table 2 molecules-28-03345-t002:** ANOVA for the response surface model.

Source	Sum of Squares	Df	Mean Squares	F Value	*p* Value
Model	87.9	9	9.77	35.78	<0.0001
Solid–liquid ratio	0.64	1	0.64	2.34	0.0317
Microwave power	2.89	1	2.89	10.59	<0.0001
Extraction time	1.19	1	1.19	4.35	0.0017
AB	2.35	1	2.35	8.6	0.0219
AC	6.18	1	6.18	22.65	0.0021
BC	0.36	1	0.36	1.32	0.2884
A^2^	17.76	1	17.76	65.06	<0.0001
B^2^	31.23	1	31.23	114.41	<0.0001
C^2^	17.67	1	17.67	64.73	<0.0001
Residual	1.91	7	0.27		
Lack of fit	1.42	3	0.47	3.85	0.1129
Pure error	0.49	4	0.12		
Cor total	89.81	16			
R2	87.9	9	9.77	35.78	<0.0001
Adj-R2	0.64	1	0.64	2.34	0.0317

**Table 3 molecules-28-03345-t003:** Box–Behnken design factors and levels.

Factor	Level		
	−1	0	1
A: Solid–liquid ratio/(g/mL)	15:1	20:1	25:1
B: Microwave power/W	300	400	500
C: Extraction time/min	45	60	75

## Data Availability

Not applicable.

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
