# Peer review of "Optimization of Extraction Process and Dynamic Changes in Triterpenoids of Lactuca indica from Different Medicinal Parts and Growth Periods"

_molecules, 2023, doi:10.3390/molecules28083345_

Round 1
Reviewer 1 Report
Dear Author,
1. What is the main question addressed by the research? The authors carried out a study to optimize the extraction conditions of triterpenes from Latuca indica, through a classic study of change of conditions, obtaining promising results on a laboratory scale and laying the foundations for the scaling up of this technique.
2. Do you consider the topic original or relevant in the field? Does it address a specific gap in the field? Optimization of any extraction method is always necessary. Personally, I searched for information in the literature on this particular topic, not finding studies that were identical or too similar to the manuscript we are discussing today, which makes me assume that there are no signs of plagiarism or that the work exactly replicates the techniques described in other previously published studies. The study is well justified, and in my very personal opinion, it fills a gap in its field of research. 3. What does it add to the subject area compared with other published material? As mentioned, I did not find studies substantially similar to the present manuscript. 4. What specific improvements should the authors consider regarding the methodology? What further controls should be considered? as I mentioned in my first review, I read the methodology described. It seems to me that the methodological approach is built according to the objectives of the study, all the trials have adequate control and the experiments are carried out with repetitions and analyzed with the necessary statistics. The results are analyzed and justified with the controls and the required statistics and are described in an orderly manner. 5. Are the conclusions consistent with the evidence and arguments presented and do they address the main question posed? Following the same argument of the previous point, the conclusions, although simple, are built and justified with the experiments carried out, there is nothing out of place, nor assumptions that are not based on evidence. 6. Are the references appropriate? I consider the bibliography adequate, there are no more comments to make about it, and surely a more exhaustive revision of the format of the bibliographical references is carried out in the editors' panel. 7. Please include any additional comments on the tables and figures. The only recommendation is to include the error bar at each point of graphics in figure 5, as they represent the mean ± standard deviation of triplicate measurements. 8. Additional comments. The manuscript can improve the use of the English language, but I do not feel qualified to point out grammatical and spelling errors.
Author Response
Please include any additional comments on the tables and figures. The only recommendation is to include the error bar at each point of graphics in figure 5, as they represent the mean ± standard deviation of triplicate measurements.
The reviewer's suggestion is very appropriate. When we looked at the figure carefully, we did not find the error bar of the data point in Figure 5. This is because the data points in the graph are too large to cover the error bars. We have recreated the graph and placed it in the manuscript.

Reviewer 2 Report
In “Figure 4 Analysis of TTC from Lactuca indica fresh samples (A-roots, C-stems, E-leaves, G-flowers, I-whole plants) and dry samples (B-roots, D-stems, F-leaves, H-flowers, J-whole plants”, there are some mistakes where is written “meddle” must be “middle”, and where is “floweing” must be “flowering”.
Author Response
In “Figure 4 Analysis of TTC from Lactuca indica fresh samples (A-roots, C-stems, E-leaves, G-flowers, I-whole plants) and dry samples (B-roots, D-stems, F-leaves, H-flowers, J-whole plants”, there are some mistakes where is written “meddle” must be “middle”, and where is “floweing” must be “flowering”.
Thank you for your careful review. The problems you found are very timely and accurate. We have remade Figure 4 and replaced it in the manuscript.

Reviewer 3 Report
Abstract and keywords: In the first sentence the authors wrote leaves, while in the second sentence they wrote different parts (….); instead of optimum should be optimal; ,,of the highest TTC parts’’ should be reformulated, line 22; ,, Leaves was the highest part of TTC in LIM.’’ should be reformulated; all investigated factors should be added in the first part of the abstract; the sentence ,,Higher TTC was extracted from fresh leaves at middle flowering stage and from dry leaves at initial flowering, and leaves were the best performing parts among the fresh and shade-dried samples, respectively.’’ is very confused and should be separated and reformulated. The last sentence should be changed. Full Latin name of the plant should be in the keywords.
Introduction: The plant should be described in more detail (botanical characteristics, usually used plant parts, chemical characterisation, its products in the market, traditional use); Soxhlet should be with uppercase letter; the sentence ,,Traditional extraction methods include reflux extraction[6] and soxhlet extraction[7], but due to the long extraction times, high temperatures and degradation of effective substances, these methods have disadvantages, including large consumption of organic solvents and low extraction efficiency.’’ should be reformulated because prolonged extraction time and high T are also disadvantages. ,,The ultrasonic extraction method uses methanol as the extraction solvent’’ is not true, there are a lot of publication where different solvents (ethanol, water, etc.) were used in UAE. Also in MAE, there were used different extraction mediums, not only ethanol.
Results and Discussion: instead of ~ should be -; In section 2.2.1 A, B, and C should be explained below the Eq.; it is very confused when the authors used only leaves and when other parts; the discussion is very weak; The sentences in lines 270-274 and 290-294 are completely unnecessary; ABTS radical neutralization results are expressed as IC50? It should be given in Material and Methods section; how can the authors claim that it is the antioxidant activity of total triterpenoids? It is the antioxidant capacity of the whole extract. The discussion of antiox results and comparison with literature data are completely missed.
Materials and Methods: Characteristics of homogeniser and lyophilizer, as well as details about lyophilisation process should be added; shade dried or lyophilized?, line 326; ,,with a volume fraction of 70% after optimization’’ – which optimization? All used power and time should be added, lines 330-331; name and producer of centrifuge should be given; writing of °C unit should be consistent; instead of minutes should be min; the authors should avoid ,,we put’’, ,,we do’’, etc; the authors did not explain determination of TTC for the extracts; check mistake, line 372; why deionized water instead of the triterpenoid solution (A0), when the extracts were ethanolic, not water?; Ai and Aj should be better explained, lines 387-388; determination of IC50 should be better explain; full meaning of SPSS should be given; HPLC analysis should be added in the study; the abbreviation of ascorbic acid cannot be VC, it can be only when author use name ,,vitamin C’’.
The English language should be improved in the whole manuscript.
The discussion is very weak or missed.
The presented study does not meet the quality criteria and impact factor of the journal Molecules.
Author Response
Responses to Review
Abstract and keywords:
- In the first sentence the authors wrote leaves, while in the second sentence they wrote different parts (….); instead of optimum should be optimal; ,,of the highest TTC parts’’ should be reformulated, line 22; Leaves was the highest part of TTC in LIM.’’ should be reformulated
Responses: According to previous studies, we found that the main utilization part of Lactuca indica L.cv. Mengzao is its leaf. We assume that the leaf of Lactuca indica L.cv. Mengzao has the most abundant flavonoids content, so we conducted optimization of the extraction process around the leaf. In order to prove our hypothesis, under the optimal technological conditions, the content of flavonoids in different parts of Lactuca indica L.cv. Mengzao were compared, and part with high flavonoid content were selected for antioxidant testing.
- all investigated factors should be added in the first part of the abstract;
Responses: Yes, we will add this section to the abstract.
- the sentence ,,Higher TTC was extracted from fresh leaves at middle flowering stage and from dry leaves at initial flowering, and leaves were the best performing parts among the fresh and shade-dried samples, respectively.’’ is very confused and should be separated and reformulated.
Responses: We have revised complex and confused sentences in the abstract.
- The last sentence should be changed. Full Latin name of the plant should be in the keywords.
Responses: We have revised and improved this part of the manuscript
Introduction:
- The plant should be described in more detail (botanical characteristics, usually used plant parts, chemical characterisation, its products in the market, traditional use);
- Responses: We have revised and improved this part of the manuscript.
- Soxhlet should be with uppercase letter; the sentence ,,Traditional extraction methods include reflux extraction[6] and soxhlet extraction[7], but due to the long extraction times, high temperatures and degradation of effective substances, these methods have disadvantages, including large consumption of organic solvents and low extraction efficiency.’’ should be reformulated because prolonged extraction time and high T are also disadvantages. ,,The ultrasonic extraction method uses methanol as the extraction solvent’’ is not true, there are a lot of publication where different solvents (ethanol, water, etc.) were used in UAE. Also in MAE, there were used different extraction mediums, not only ethanol.
Responses: Thank you for the professional introduction and correction provided by the reviewer. We carefully reviewed the relevant contents and found that the introduction of ultrasound-assisted extraction and microwave-assisted extraction was inappropriate. We have revised and improved this part of the manuscript.
Results and Discussion:
- instead of ~ should be -; In section 2.2.1 A, B, and C should be explained below the Eq.;
Responses: We have already described this part in detail.
- it is very confused when the authors used only leaves and when other parts; the discussion is very weak; The sentences in lines 270-274 and 290-294 are completely unnecessary;
Responses: We have supplemented and modified the content you mentioned, as well as modified complex sentences. After our careful consideration, we feel that these contents are of great significance for the interpretation of the article.
- ABTS radical neutralization results are expressed as IC50? It should be given in Material and Methods section;
Responses: In vitro antioxidant testing, IC50 values were selected to indicate the antioxidant activity of different antioxidants. The IC50 value represents the semi inhibitory concentration. The lower the IC50 value, the stronger the free radical scavenging ability of the antioxidant. In addition, in the materials and methods section, we have added an explanation of IC50.
- how can the authors claim that it is the antioxidant activity of total triterpenoids? It is the antioxidant capacity of the whole extract. The discussion of antiox results and comparison with literature data are completely missed.
Responses: We strongly agreed with the suggestions of the reviewers. In this study, triterpenoids were not isolated, so our research object was the MAE extracts from the leaves of Lactuca indica L.cv. Mengzao. At present, there are few studies on the antioxidant capacity of Lactuca indica. Our research results have enriched the data on the antioxidant capacity of Lactuca indica resources. In the next step, we will conduct in-depth research on the mechanism of antioxidant capacity.
Materials and Methods:
- Characteristics of homogeniser and lyophilizer, as well as details about lyophilisation process should be added; shade dried or lyophilized?,
Responses: We added the characteristics of homogeniser and lyophilizer, as well as details about lyophilisation process in the manuscript. The drying method of Lactuca indica L.cv. Mengzao samples is to use vacuum freeze drying.
- line 326; ,,with a volume fraction of 70% after optimization’’ – which optimization? All used power and time should be added,
Responses: We carefully checked the experimental process and gave a correct description.
- lines 330-331; name and producer of centrifuge should be given; writing of °C unit should be consistent; instead of minutes should be min;
Responses: Name and producer of centrifuge have been given in the manuscript. The format of the unit has also been revised
- the authors should avoid ,,we put’’, ,,we do’’, etc;
Responses: We have modified the relevant incorrect language.
- the authors did not explain determination of TTC for the extracts;
Responses: In the manuscript, we have added the determination of TTC for the extracts.
- check mistake, line 372; why deionized water instead of the triterpenoid solution (A0), when the extracts were ethanolic, not water?; Ai and Aj should be better explained
Responses: The antioxidant properties we investigated are the sample extracts, instead of triterpenoid extracts. The explained results of Ai, Aj, and AO are presented in the manuscript.
- lines 387-388; determination of IC50 should be better explain; full meaning of SPSS should be given; HPLC analysis should be added in the study; the abbreviation of ascorbic acid cannot be VC, it can be only when author use name ,,vitamin C’’.
Responses: The IC50 is better explained in the revised manuscript. Full meaning of SPSS is the Statistical Product and Service Solutions. In the next study, we will conduct HPLC analysis of triterpenoids. We replaced Vc with L-Ascorbic acid.
- The English language should be improved in the whole manuscript.
Responses: In the revised manuscript, the English language of the full text was modified.

Round 2
Reviewer 3 Report
Abstract: the sentence ''The solid–liquid ratio, microwave power and extraction time were employed effects.'' should be reformulated; after freeze -drying OF the samples.
Introduction: one sentence about ultrasound extraction should be given.
Results and Discussion: The authors did not improve the Discussion section, thus the discussion is still very weak.
Author Response
Abstract: the sentence ''The solid–liquid ratio, microwave power and extraction time were employed effects.'' should be reformulated; after freeze -drying OF the samples.
Response: The author has corrected the content you mentioned
Introduction: one sentence about ultrasound extraction should be given.
Response: In recent years, there are many advanced extraction methods on the market, such as microwave-assisted extraction, ultrasound-assisted extraction and supercritical fluid extraction. Each of these extraction methods has certain advantages. In this study, the main purpose of the author is to optimize the microwave-assisted condition. Therefore, there is relatively little talk about ultrasound extraction.
Results and Discussion: The authors did not improve the Discussion section, thus the discussion is still very weak.
Response: According to the suggestions of reviewers, the author has effectively supplemented and modified the content of the manuscript.